# EBV dUTPase: A Novel Modulator of Inflammation and the Tumor Microenvironment in EBV-Associated Malignancies

**DOI:** 10.3390/cancers15030855

**Published:** 2023-01-30

**Authors:** Marshall V. Williams, Irene Mena-Palomo, Brandon Cox, Maria Eugenia Ariza

**Affiliations:** 1Department of Cancer Biology and Genetics (CBG), The Ohio State University Wexner Medical Center, Columbus, OH 43210, USA; 2Institute for Behavioral Medicine Research (IBMR), The Ohio State University Wexner Medical Center, Columbus, OH 43210, USA

**Keywords:** herpesviruses, Epstein–Barr virus, pre-latent phase, deoxyuridine triphosphate nucleotidohydrolase (dUTPase), tumor microenvironment (TME), germinal center (GC) reaction

## Abstract

**Simple Summary:**

In this review, we summarize the current knowledge concerning the potential roles of “abortive-lytic” replication of EBV in establishing the latent state and its contribution to the tumor microenvironment (TME) and oncogenesis. Particular emphasis is given to discussing how EBV deoxyuridine triphosphate nucleotidohydrolase (dUTPase), an early protein encoded by BLLF3 which possesses novel immunomodulatory properties, may contribute to these processes.

**Abstract:**

There is increasing evidence that put into question the classical dogma that the Epstein–Barr virus (EBV) exists in cells as either a lytic virus in which new progeny is produced or in a latent state in which no progeny is produced. Notably, a third state has now been described, known as the abortive-lytic phase, which is characterized by the expression of some immediate early (IE) and early (E) genes, but no new virus progeny is produced. While the function of these IE and E gene products is not well understood, several recent studies support the concept they may contribute to tumor promotion by altering the tumor microenvironment (TME). The mechanisms by which these viral gene products may contribute to tumorigenesis remain unclear; however, it has been proposed that some of them promote cellular growth, immune evasion, and/or inhibit apoptosis. One of these EBV early gene products is the deoxyuridine triphosphate nucleotidohydrolase (dUTPase) encoded by BLLF3, which not only contributes to the establishment of latency through the production of activin A and IL-21, but it may also alter the TME, thus promoting oncogenesis.

## 1. Introduction

The Epstein–Barr virus (EBV), a γ-herpesvirus, infects a significant percentage of the adult population (95%) worldwide. Primary infections in developed countries usually occurs in adolescents and is generally asymptomatic except for a small percentage of individuals who develop infectious mononucleosis [1,2]. EBV establishes a persistent infection in memory B cells and can be reactivated to undergo lytic replication often during an individual’s lifetime [3,4,5,6].

EBV has the distinction of being the first human oncogenic virus identified [7]. EBV has been implicated in the pathogenesis of several human malignancies and lymphoproliferative disorders in immunocompetent and immunosuppressed individuals [2]. These malignancies include nasopharyngeal carcinoma (NPC), gastric carcinoma (GC), Hodgkin lymphoma (HL), Burkitt lymphoma (BL), diffuse large B-cell lymphoma (DLBCL), as well as extranodal NK/T-cell lymphoma, and nasal type (ENKT-NT) (Figure 1). Wong et al. estimated EBV-related malignancies accounted for 239,700–357,900 new cases and 137,900–208,700 deaths in 2020, approximately 1.3–1.9% of the global cancer burden [8].

In recent years, the concept that EBV may exist in two phases, the lytic phase in which new virus progeny is produced and latent phase in which one of four latency gene expression programs are expressed but no virus progeny is produced, may be incorrect. A third phase “abortive-lytic replication” has been identified in which specific immediate early (IE) and early (E) genes associated with EBV lytic replication are expressed but no virus progeny is produced [9,10,11,12,13]. Furthermore, examination of tissue from various EBV-related malignancies has demonstrated EBV genes associated with “lytic” replication are expressed, thus supporting the hypothesis that EBV encoded lytic proteins may be contributing to tumor initiation/promotion [14,15,16,17,18,19,20,21,22].

In this review, we summarize the current knowledge concerning the potential roles of “abortive-lytic” replication of EBV in establishing the latent state and its contribution to the tumor microenvironment (TME) and oncogenesis. Particular emphasis is given to discussing how EBV deoxyuridine triphosphate nucleotidohydrolase (dUTPase), an E protein encoded by BLLF3, which possesses novel immunomodulatory properties, may contribute to these processes.

## 2. Establishment of Latency

Following oral transmission of EBV, the virus infects epithelial cells, as well as naïve mucosal B cells, ultimately establishing latency in memory B cells [3,4,5,6]. Thorley-Lawson proposed the germinal-center (GC) model to explain the mechanism(s) by which EBV establishes the latent state in memory B cells in immunocompetent individuals, and it is the most widely accepted model at this time [23]. According to this model, EBV uses a sequential expression of four virus-encoded latency programs (Latency III, II, I and 0) to guide infected resting naïve B cells to become proliferating blasts, participate in GC reactions, and to enter the resting memory B-cell (MBC) compartment. EBV remains latent in MBCs until it differentiates into a plasma cell at which time EBV re-enters the lytic/abortive-lytic replication cycle.

Over the years, there has been great interest in determining what happens in the early events following the infection of a naïve B cell by EBV. While a challenging task, the development of new technologies, such as single cell transcriptome analyses, have made possible to study and characterize gene expression changes of B cells, as well as changes in the expression of EBV genes in B cells infected with EBV in vitro.

### 2.1. Pre-Latent Phase

Upon infection of naïve B cells and the release of the virion DNA into the nucleus, the virion DNA, which lacks nucleosomes and is unmethylated, is rapidly chromatinized by histone assembly and CpG methylated. This process has been estimated to last from eight days to several weeks, and it is referred to as the pre-latent phase [24,25]. Several studies have now demonstrated global changes in the transcriptome of infected B cells are required to establish a successful latent infection [13,26,27]. Interestingly, during the pre-latent phase, EBV uses this early time window to express a limited set of viral genes belonging to the immediate-early (IE) and early (E) classes of lytic viral genes [28,29,30,31], as well as some microRNAs [32], and other genes associated with latency [33,34] (Figure 2).

Interestingly and somewhat unexpected, several of these pre-latent phase genes are essential or support initial steps required for the establishment of latent infections in primary human B cells [35]. While some initial studies suggested a productive infection also occurred in these newly infected naïve B cells [29], a recent study has recently demonstrated rather conclusively this is not the case; rather, this is an abortive-lytic infection [13]. The molecular mechanism(s) regulating/inducing this pre-latent lytic gene expression is still elusive, but it may be accounted for by uncontrolled gene expression from the EBV genome, which lacks suppressive epigenetic modifications, the release of tegument proteins following the uncoating process, and/or by viral mRNAs incorporated into the virion that are released upon infection [36]. In the pre-latent phase of viral infection, EBV reprograms the resting B cell into activated proliferating B blasts [27,36]. While the functions of these lytic proteins in the establishment of the latency phase remain unclear, it has been proposed to increase proliferation of the infected B cell, prevent apoptosis, and act as immune evasion mechanisms (Table 1). The E gene, BLLF3, which encodes for the deoxyuridine triphosphate nucleotidohydrolase (dUTPase), has recently been demonstrated to be expressed during the pre-latent phase [37]. Interestingly, the EBV dUTPase protein has been shown to elicit novel immune functions in addition to its well-known enzymatic activity [38,39,40,41,42,43,44,45,46]. Potential activities of the dUTPase in the pre-latent phase are depicted in Figure 3.

BZLF1 encodes for Zebra (Zta), an IE protein and important transcriptional regulator. BZLF1 is expressed during the pre-latent phase and regulates the switch from lytic to latent phase. It accomplishes this by binding to two classes of ZEBRA response elements (ZREs): (1) CpG-free motifs resembling the consensus AP-1 site recognized by cellular bZIP proteins and (2) CpG-containing motifs that are selectively bound by ZEBRA upon cytosine methylation [47,48]. There have been numerous reports describing the function of BZLF1 as the molecular switch regulating the transition between lytic and latent phases of EBV [24,48,49] and how Zta promotes oncogenesis [50,51,52].

The promoter sequence of BLLF3 contains a CpG-free motif, and thus, its expression is not subjected to epigenetic regulation. This would be necessary if the dUTPase protein had a functional role in the initiation/maintenance of latency [53]. In support of this premise, a recent study by Cox et al. [46] reported the EBV dUTPase induces the secretion of activin A, a pleiotropic cytokine and potent inducer of proliferation and differentiation of IL-21 producing follicular CD4^+^ helper T cells (T_FH_) [54]. T_FH_ cells are specialized providers of T-cell help to B cells and are essential for GC formation, affinity maturation, and the development of most high-affinity antibodies and memory B cells [55].

Activin A belongs to the TGF-β superfamily and can affect several cell types involved with immune regulation. Activin A has been reported to have dual and opposite roles. It exerts an oncogenic role in head and neck squamous cell carcinomas in which activin A expression is correlated with increased proliferation, invasion, and poor patient prognosis but also has tumor suppressor roles in prostate and breast cancers [56]. Studies on haemopoietic malignancies are more limited, but Portale et al. [57] reported activin A confers a migratory advantage to acute lymphoblastic leukemia cells.

Interleukin-21 (IL-21) is a pleiotropic cytokine that has diverse effects on numerous cell types, including those involved with immune function [58,59,60,61,62]. IL-21 has been employed in multiple clinical studies for the treatment of various malignancies, particularly metastatic renal cell carcinoma and metastatic refractory non-Hodgkin lymphoma [63,64,65,66]. In vivo and in vitro studies have demonstrated IL-21 exerts diverse regulatory effects on healthy and tumor cells depending on the type of cell, stage of differentiation, stimuli, and EBV status. IL-21 induces B-cell proliferation and sustains normal GC reactions following appropriate B-cell receptor (BCR) signaling and a T-cell dependent response. These pro-survival properties of IL-21 may have a role during the initial stages of GC-derived malignant transformation. However, in more advanced malignancy stages, such as diffuse large B-cell lymphoma (DLBCL), IL-21 has been shown to exhibit anti-cancer activity by downregulating anti-apoptotic genes and promoting apoptosis, as well as growth arrest [64,67,68,69,70,71,72,73]. IL-21 has also been suggested [74] to exhibit anti-tumor properties in lymphomas by expanding and enhancing tumor-infiltrating cytotoxic CD8^+^ T cells and NK cells [75,76,77,78].

### 2.2. What Is the Relationship if Any between IL-21 and EBV?

There have been few in vitro studies examining the effect of IL-21 on EBV gene expression. Using early-passage EBV infected synovial tissue derived B-cell lines (OCI-BCLs), which exhibit latency type III, Konforte et al. [79] reported IL-21 initially decreased the constitutive expression of the IE gene, BRLF1, which encodes for the transcriptional activator, Rta. However, this was followed by an increased in BZLF1, BRLF1, and BMLF1 expression, possibly due to the differentiation of OCI-BCLs to immunoglobulin (Ig)-secreting late plasmablasts/early plasma cells [80]. A follow up study by this group revealed IL-21 decreased gene and protein expression of EBV nuclear antigen 2 (EBNA2) and upregulated the expression of LMP1 in OCI-BCLs [81]. Furthermore, studies by Kis et al. [82] showed the effects of IL-21 were dependent on the type of EBV latency gene program being expressed. In some cell lines (BL and the lymphoblastoid cell line (LCL) ER/EB2-5) expressing the latency type I program, IL-21 induced the expression of LMP1 but not EBNA2, while in others (Jijoye), cell proliferation was inhibited. In type III LCLs and BL cell lines, IL-21 repressed LMP-2A and up-regulated the expression of LMP-1 mRNAs. The IL-21-treated type III cells also underwent plasma cell differentiation. However, since the concentrations of IL-21 used in these studies (50 and 100 ng/mL) were approximately 500 to 1000-times the concentration found in sera of healthy individuals [46], it remains to be determined whether IL-21 has any effects on EBV gene expression in vivo.

## 3. EBV-Associated Germinal Center Malignancies

The germinal center (GC) is a specialized microstructure that forms in secondary and tertiary tissue, where B cells are programmed to become memory B cells and high-affinity antibody-producing plasma cells in a complex process that requires B cells to undergo a high rate of cell division and antigen-driven somatic hypermutation (SHM). Mutations during this process may have an unfavorable outcome driving lymphomagenesis [83,84,85]. In fact, GCs are the origin of follicular lymphoma (FL), GC-diffuse large B-cell lymphoma (DLBCL), and Burkitt lymphoma (BL). Notably, EBV has been implicated in each of these malignancies. EBV-positive FL is an uncommon and poorly characterized disease identified in 2.5% of all FL cases, mostly with a type II latency program [86]. FL patients have an increased risk of transforming into an aggressive and refractory form of DLBCL [85]. The oncogenic role of EBV in leading the transformation of FL to a high-grade form is still controversial [87] but appears to be somewhat similar to events that occur in some patients with chronic lymphocytic leukemia in Richter’s transformation [88]. These studies identify a critical research gap, which calls for follow-up investigations focused on examining the expression pattern of EBV lytic genes in FL patients and those undergoing transformation to an aggressive form of DLBCL.

### 3.1. EBV-Positive Diffuse Large B-Cell Lymphoma (EBV^+^-DLBCL)

EBV-DLBCL constitutes a distinct clinicopathological entity in the World Health Organization (WHO) classification [89]. The incidence varies from 2–19%, and the overall prognosis of this malignancy remains unclear [90]. While genomic studies of EBV^+^-DLBCL are limited [91,92,93], it is well established that infection of naïve B cells by EBV results in major alteration in the expression of host genes, as well as the regulated expression of EBV genes. In vitro studies employing primary cells from EBV positive DLBCL tumor cells and cell lines revealed EBV prevented IL-21-induced apoptosis [94], and EBV infection of B cells provided survival factors to EBV^+^-DLBCL cell lines and modulated cytokine-induced specific chemotaxis in these cells [95]. Notably, Wang et al. [96] recently reported that IL-21 stimulated the expression and activation of cell cycle regulators and promoted cell proliferation of EBV^+^-DLBCL primary cells and cell lines. Recent studies by others also demonstrated the expression of lytic genes, BZLF1, BHRF1, BLLF1, [97] and BLLF3 [98] in EBV^+^-DLBCL cells and the presence of anti-EBV dUTPase (BLLF3) antibodies in serum of patients with EBV^+^-DLBCL [44]. Interestingly, the effects of IL-21 in EBV^+^-DLBCL are not observed in EBV-negative DLBCL, suggesting a role for EBV lytic genes/products as contributors to the TME while highlighting the need of using distinct therapeutic approaches for the treatment of EBV-positive versus EBV-negative DLBCL.

### 3.2. Classical Hodgkin Lymphoma (cHL)

EBV is associated with a subset of patients with cHL. cHL is defined by the presence of Hodgkin and Reed–Sternberg (HRS) cells within a background of reactive cells primarily composed of B and T lymphocytes, plasma cells, macrophages, and eosinophils. HRS cells develop from matured B cells that have undergone somatic hypermutation, suggesting they are derived from GC or post-GC B cells. Notably, infectious mononucleosis caused by EBV confers an increased risk for the development of EBV-positive cHL. More importantly, evidence of high-antibody levels against EBV viral capsid (VCA) and early lytic antigens are considered risk factors along with immune suppression and/or immune senescence. Additionally, a recent study demonstrated the BLLF3 gene was expressed in 67% (*n* = 3) of cHL biopsies analyzed [94]. How EBV contributes to cHL initiation/progression remains unclear, but it has been suggested EBV may be partly responsible for reshaping the TME through expression of latent genes and limited induction of the lytic cycle [99,100].

### 3.3. Burkitt Lymphoma (BL)

BL is an aggressive B-cell non-Hodgkin lymphoma first described in Africa and caused by EBV [7]. BL is divided into three main clinical variants: endemic, sporadic, and immunodeficiency associated. Patients with the endemic variant of BL exhibit the greatest positivity for EBV (>90%) and latency I or Wp-restricted gene expression program [101]. The genetic hallmark of all three BL types is the chromosomal translocation of *myc* gene to one of the three immunoglobulin loci. However, studies in mice and humans have shown deregulation of *MYC* alone is not sufficient to drive BL lymphomagenesis. Genomic profiling studies have focused primarily on identifying host genes that may act as drivers of BL, and only one study has described the expression of genes involved with EBV lytic replication in BL. RNA sequencing of endemic BL tissue (*n* = 26) detected the expression of EBV lytic transcripts for BILF1, BALF4, and LF2 in all 26 cases, and of BALF2 (90%), BHRF1(80%), BZLF1 and BMRF1 (60%), BNLF2a (50%), and BCRF-1 (45%) [102]. The role(s) of these lytic proteins in the initiation and/or promotion of BL is unknown.

### 3.4. T_FH_ Derived Lymphomas

Angioimmunoblastic T-cell lymphoma (AITL) and T_FH_-type peripheral T-cell lymphoma (T_FH_-PTCL) are T_FH_ cell derived lymphomas. EBV is known to be associated with the pathogenesis and histological progression of AITL. AITL is probably the most common peripheral T-cell lymphoma in Asian populations, and it has a poor prognosis. EBV is not often found in neoplastic T cells but rather in adjacent B cells. A recent transcriptomic study of 14 clinical AITL samples demonstrated that while an EBV latency type II expression pattern was observed, all samples expressed some lytic genes, but no virus replication occurred, suggesting abortive-lytic replication [98]. Interestingly, 86% (12/14) of the AITL samples examined expressed BLLF3, which encodes for the dUTPase protein.

Peripheral T-cell lymphoma (PTCL) comprises a heterogenous group of uncommon lymphomas derived from mature, post-thymic or “peripheral” T and natural killer cells, and T_FH_-PTCL. The significance of EBV infections in T_FH_-PTCL remains unclear. Most genetic profiling studies have focused on identifying mutations in host genes and few studies have examined the expression of EBV genes. A recent study by Nakhoul et al. [103] reported the expression of several EBV E genes, including BLLF3, in AITL tumors but no BLLF3 transcripts were detected in PTCL, not otherwise specified (PTCL-NOS) (*n* = 10), while another study by Bayda et al. [98] detected BLLF3 transcripts in two of three tumor samples examined. Obviously, further studies are needed to ascertain what EBV genes encoding for lytic proteins are expressed in these lymphoma types and the potential role of the dUTPase and other lytic proteins in the development of these lymphomas.

Altogether, these results suggest the expression of BLLF3 and subsequent production/secretion of the dUTPase protein during the pre-latent phase may prime the formation of T_FH_ cells to aid in the establishment of EBV latency according to the GC model. It would also suggest mutations affecting the functions of EBV dUTPase would negatively affect the establishment/maintenance of EBV latency. Furthermore, the data from these studies demonstrate that by stimulating T_FH_ to produce IL-21, the dUTPase has the potential to alter the TME by enhancing the survival and proliferation of EBV positive DLBCL NOS cells. Finally, while BLLF3 expression has been reported in other EBV-related lymphomas, such as AITL, PTCL, and BL, its potential role in tumor initiation/promotion in these malignancies remains unknown.

## 4. Modulation of the Tumor Microenvironment (TME)

### 4.1. By EBV dUTPase

The TME contains in addition to malignant cells, a heterogeneous collection of infiltrating immune cells, including B, T, and NK cells, macrophages, dendritic cells, and neutrophils, as well as resident host cells, secreted factors, and extracellular matrix. The TME is a vital component of many neoplastic diseases, including lymphomas. However, the TME of lymphomas vary greatly. In Hodgkin lymphomas (HL) as well as several T-cell lymphoma entities, such as AITL, greater than 80% of the tumor mass consists of TME cells, while in FL the TME constitutes about 50% of the cellular mass. In DLBCL, the proportion of the TME varies and is generally lower, while in BL, plasmablastic lymphoma and lymphoblastic T-cell and B-cell lymphomas, the TME is barely existent. Interactions between lymphoma cells and the TME are important for the survival and proliferation of lymphoma cells, which reprogram the TME to protect them from the host’s immune system defense mechanisms. The immunomodulatory/immunosuppressive role of the TME is of interest in the hopes of identifying new targeted therapies. The development of EBV-associated malignancies is tightly associated with the TME and shaped by tumor cells to suppress the host’s immune system and evade immune surveillance [104,105,106,107]. The possible roles of the dUTPase protein in altering the TME are shown in Figure 4.

### 4.2. Exosomes

Exosomes are membrane bound vesicles 40–100 nm in size that are produced through an exosomal pathway. During their formation, they incorporate as cargo, virus, and host cell proteins, DNA, messenger RNAs (mRNAs), and microRNAs (miRNAs). The composition of the exosomal cargo varies depending on the host cell in which they are produced and may have immune stimulatory, inhibitory, or tolerance-inducing effects. Exosomes have been implicated in numerous diseases and are capable of trafficking to various organs within the body, where they function as intercellular messengers. Several studies have described the potential roles of endosomes in oncogenesis [108].

EBV-associated exosomes have been reported to contain various EBV products, including LMP1, LMP2, EBV-encoded small RNAs (EBERs), mRNAs encoding LMP1, LMP2, EBNA1 and EBNA2, miRNAs, the late lytic proteins, BGLF2 and gp350, and various host cell products. There is a significant body of literature demonstrating EBV gene products in exosomes can cause immune activation through several signaling pathways, resulting in the induction of type 1 interferons, pro-inflammatory cytokines, and cell surface receptors [109,110,111,112,113,114,115,116], and to either inhibit or facilitate EBV infection of other cells [117,118]. The EBV dUTPase protein has also been shown to be released in exosomes. Exosomes containing EBV dUTPase protein have been reported to induce NF-κB activation and cytokine secretion (IL-1β, IL-6, IL-8, IL-10, IL-12p70, TNF-α and IFN-γ) in primary human DCs and PBMCs through a Toll-like receptor 2 (TLR2) mechanism [39,40,41,42] and to convert phagocytes into tumor-associated macrophages (TAMs) via induction of the inflammatory response [119]. Altogether, these data demonstrate EBV dUTPase protein is packaged into exosomes in vitro where it can elicit diverse effects, ranging from increasing infectivity to modifying the TME.

### 4.3. Inflammation: Cytokines

Around 15% to 20% of all cancer cases are preceded by infection, chronic inflammation, or autoimmunity [120,121]. It is well documented that inflammation is induced and exists long before tumor formation occurs. In fact, EBV^+^ DLBCL develops in a setting of longstanding chronic inflammation due to infectious/autoimmune conditions [122]. Inflammatory processes have also been shown to be involved not only with promotion but also survival of the tumor [123]. Cytokines are families of secreted molecules which play a central role in cell signaling, cell-to-cell communication, and inflammation. The expression of several pro-inflammatory cytokines, IL-1β, IL-6, IFN-γ, and TNF-α, as well as the chemokine, IL-8, are known to be upregulated in EBV-associated malignancies [124,125,126,127,128,129]. In addition, IL-10, an anti-inflammatory cytokine [130], has also been shown to be upregulated in EBV-associated malignancies. While most studies focused on the role of the latency-expressed protein, LMP1, in cytokine induction [114,131], the lytic EBV dUTPase protein has been reported to induce IL-1β, IL-6, IL-8, TNF-α, INF-γ, and IL-10 through a TLR2-dependent mechanism that leads to the activation of NF-κB [39,40,41,42]. The interplay between these different cytokines, chemokines, and interferons regulate the growth of malignant cells and may alter EBV gene expression directly or indirectly [132,133].

### 4.4. Checkpoint Molecules

Immunomodulatory molecules, such as programmed cell death protein 1 (PD-1), and its ligands-programmed cell death protein ligand 1 and 2 (PD-L1 and PD-L2), play important roles in assisting tumor cells to escape the host immune system [132,134]. LMP1 has been reported to upregulate the expression of PD-L1 in EBV-infected cells through activation of TLR signaling and signal transduction and activation of transcription 3 (STAT3) transcription factor or through the engagement of the activator protein-1 (AP-1) associated enhancers [135,136]. Likewise, EBV miRNAs have been shown to modulate the expression of PD-L1 and 2 [137,138]. Notably, the EBV dUTPase protein has been reported to inhibit T-cell proliferation in vitro following anti-CD3 stimulation [39]. Microarray analyses of human dendritic cells have also demonstrated the EBV dUTPase protein-modulated pathways either positively (PD-1: PD-L1/L2; inducible costimulatory molecule—ICOS: ICOS ligand—ICOSL) or negatively (PAG1: phosphoprotein associated with glycosphingolipid-enriched microdomains 1) that could promote T-cell tolerance/exhaustion [42,43]. Furthermore, the EBV dUTPase protein has been shown to induce the expression of B-cell integration cluster (BIC) transcript, the precursor of miR-155 [42]. MiR-155 is a multifunctional micro-RNA enriched in cells of the immune system and is indispensable for the immune response. However, when dysregulated miRNA-155 contributes to the development of chronic inflammation, autoimmunity, cancer, and fibrosis [139]. These results suggest the EBV dUTPase protein may have an important role in promoting immune avoidance in EBV-associated malignancies.

## 5. Other EBV-Associated Malignancies

### 5.1. Nasopharyngeal Carcinoma (NPC)

NPC is a rare type of head and neck epithelial cancer that originates in the nasopharynx. Most endemic NPC cases are classified as undifferentiated non-keratinizing WHO type-III tumors, and EBV is implicated as the causal agent in most cases. While NPC is a latency II associated cancer, there is new evidence supporting an important role for proteins produced during abortive-lytic replication in the initiation and/or progression of NPC. Notably, abortive-lytic replication of EBV has been reported in several studies [140,141,142,143]. In fact, deep sequencing of NPC-derived C666-1 cells revealed the E genes, BBLF4, BGLF4, BHRF1, and BLLF3, as well as the L gene BGLF2, were expressed in all the EBV-positive cells [144]. Furthermore, in a study using a severe combined immunodeficient (SCID) mouse-C666-1 tumor model, it was reported that several EBV lytic genes, including BLLF3 encoding the dUTPase protein, were expressed. These results along with findings of a previous study demonstrating the presence of anti-EBV dUTPase antibodies in 44% of NPC cases examined (*n* = 16) when compared to healthy controls (4.6%, *n* = 160) [43], further support the premise that EBV dUTPase is expressed in NPC malignancies and may contribute to NPC pathophysiology.

### 5.2. Gastric Carcinoma

Gastric cancer is the fifth most common malignant tumor and second leading cause of cancer-related deaths worldwide. It has been reported that EBV is a potential etiological agent in approximately 10% of gastric cancers. However, EBV’s role in gastric carcinogenesis is poorly understood [145]. Several studies reported the expression of lytic EBV genes in gastric tumor tissue (*n* = 25), including the IE genes BZRF1 and BRLF1 [146,147]. Using a cohort of gastric carcinoma RNA-seq data sets from The Cancer Genome Atlas (TCGA), Strong et al. [148] conducted a quantitative and global assessment of EBV gene expression in gastric carcinomas and examined EBV-associated cellular pathway alterations. They found that in addition to LMP1 and LMP2, IE genes, BZLF1 and BRLF1, were also expressed but no progression beyond E gene expression occurred, thus suggesting abortive-lytic replication in vivo. Notably, Borozan et al. [149] identified multiple lytic genes (IE: BZLF1, BRZF1; E: BALF5, LF2, BNLF2a, BNLF2b, BILF, LF1, BALF3, BARF1, BALF2, BALF1, BKRF4, BKRF3; L: LF3, BALF4, BNRF1, BPLF1) in eight gastric cancer tissues. They concluded the expression of these genes was not consistent with typical lytic or abortive-lytic signature and suggested this may represent novel mechanisms to activate the expression of some EBV lytic proteins and contribute to oncogenesis. Lastly, Song et al. [150] reported the identification of nine IgG antibodies that were discriminative for tumor EBV status. These antibodies included anti-LF2, anti-BRLF1, anti-BORF2 (inhibits cellular APOBEC3B), anti-BALF2, anti-BaRF1, anti-BXLF1, and anti-BLLF3. Interestingly, these investigators concluded the EBV-associated gastric carcinoma-specific humoral response was exclusively directed against lytic cycle immediate-early and early antigens, unlike other EBV-associated malignancies.

### 5.3. Extranodal NK/T-Cell Lymphoma, Nasal Type (ENKT-NT)

ENKT-NT, which is considered the prototype of EBV-driven T and NK cell lymphoproliferative disorders, is seen primarily in Eastern Asia and Latin America but is rare in the United States and Europe [151,152]. A molecular classification has been recently proposed distinguishing three different ENKT-NT subtypes defined according to their genetic characteristics [153]. The subtypes are: TSIM (exhibit alterations in tumor suppressors and immune modulators), MB (exhibit mutations in the tumor suppressor gene MGA), and HEA (have mutations in histone deacetylaseHDAC1, histone acetyltransferase EP300, and ARID1A, a tumor suppressor gene involved in chromatin remodeling). When EBV transcripts were investigated among the three molecular subtypes, different patterns were observed. The TSIM subtype exhibited a latency type II gene expression profile and the greatest expression of BALF3, an E-expressed lytic gene. The MB subtype exhibited a latency type I-like gene expression profile, while the HEA subtype exhibited a latency type II gene expression profile and the highest expression level of BNRF1, an L-expressed lytic gene. Examination of somatic single-nucleotide variations (SNVs) of EBV genome were also strongly associated with disease pathogenesis [154]. Interestingly, one SNV that occurred in 97% of the tumors examined was in BLLF3 (S253G) [153]. Similar findings were reported by Peng et al. [154]. The results from these studies potentially implicate EBV dUTPase in ENKT-NT; however, the specific role of EBV dUTPase in ENKT-NT is largely unknown.

## 6. Autoimmune Disease and Lymphoma

Autoimmune diseases constitute a clinically heterogeneous group of disorders, affecting up to 10% of the population worldwide. Autoimmune diseases include inflammatory bowel disease (IBD), multiple sclerosis (MS), type 1 diabetes (T1D), systemic lupus erythematosus (SLE), rheumatoid arthritis (RA), ankylosing spondylitis (AS), Sjogren syndrome (SS), as well as psoriasis and psoriatic arthritis (PPA). While there is an association between autoimmune disease and cancer development, the strength of the association as well as the inherent risk varies between different autoimmune diseases, different types of malignancies, and different populations [155,156,157]. Autoimmune diseases are an established risk factor for lymphoma, conferring a two- to thirty-seven-fold increased risk. However, their potential role in the initiation/promotion of various lymphomas remain unknown [158,159].

EBV has also been implicated is several autoimmune diseases, including MS, SLE, RA, and SS [160,161,162,163,164]. However, except for a single study which demonstrated the presence of EBV dUTPase in plasma cell infiltrates in kidneys of lupus nephritis patients [165], there have not been any studies to address the potential role of EBV-dUTPase in other autoimmune diseases.

Molecular mimicry is considered by many to be the primary mechanism by which infectious agents, such as EBV, may induce autoimmunity [166]. Studies involving EBV have implicated EBNA1 in MS [167,168,169] and EBNA1 in conjunction with LMP-1 in SLE [170] through a molecular mimicry mechanism.

Another potential mechanism that has received little attention in EBV infections is the extrafollicular antibody response. Following a viral infection, activated B cells may undergo a GC or an extrafollicular response. While a GC response typically results in the production of high affinity antibodies that contribute to long-term protection, extrafollicular responses generally produce lower affinity antibodies against pathogens [171]. Interestingly, a recent study reported, using a mouse model, that the EBV dUTPase protein stimulated an extrafollicular antibody response, as determined by the significant increase in the frequency of invariant natural killer T (iNKT) cells, marginal zone (MZ) B cells, and plasmablasts/plasma cells in vivo [46]. In addition to the development of extrafollicular loci and abortive GC formation, the interaction of iNKT cells with B cells can lead to short-lived low-affinity plasma cells. Additionally, MZ B cells can differentiate into extrafollicular low-affinity plasma cells [172,173]. While these processes result in a rapid antibody response to viruses, dysregulation of either can result in autoantibody production. Whether this process contributes to lymphoproliferative disorders or not is unknown.

## 7. Summary and Future Directions

Studies concerning the mechanism(s) by which EBV contributes to oncogenesis have focused extensively on EBV gene products expressed during the latent phase, especially LMP1 and LMP2A, even when early studies reported the presence of proteins expressed during the lytic phase. Over the past 20 years the concept that EBV exists in vivo in either a lytic or latent phase has been questioned, and recent studies using more advanced technologies have demonstrated the presence of a third in vivo phase known as the abortive-lytic phase. This abortive-lytic phase is characterized by the expression of some but not all lytic EBV genes. During the lytic phase of EBV replication, approximately 80 genes are expressed in a coordinated cascade resulting in the production of progeny. This is not the case during the abortive-lytic phase as only a few of these lytic genes are expressed. One such gene is BLLF3, which encodes for a dUTPase. The EBV dUTPase protein possesses, in addition to its classical enzymatic activity, novel properties based upon its function as a pathogen-associated molecular pattern ligand for TLR2. There is accumulating evidence supporting the dUTPase protein as playing an important role in initiating/maintaining the latency phase and simultaneously contributing to oncogenesis. However, additional studies are needed, which focus on examining the expression of this gene (BLLF3) in malignant tissues and to elucidate the potential role(s) of the dUTPase protein in modulating oncogenesis.

## Figures and Tables

**Figure 1 cancers-15-00855-f001:**
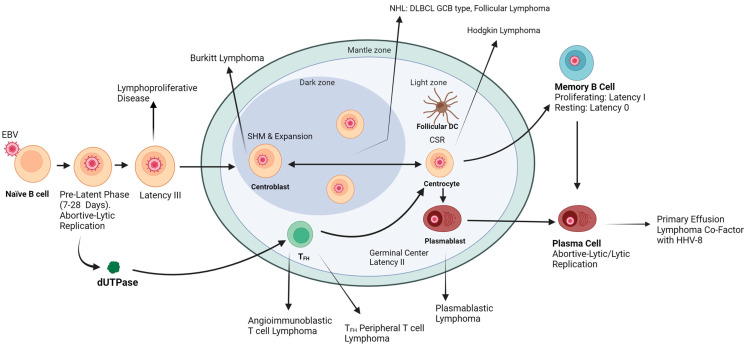
Illustration of germinal center (GC) reaction indicating the events required for affinity maturation of antibody responses, the potential role of the EBV dUTPase protein in this process, and the cellular origins of various EBV-associated B- and T-cell lymphomas. DC = dendritic cell; SHN = somatic hypermutation; CSR = class-switch recombination; T_FH_ = follicular CD4^+^ helper T cell.

**Figure 2 cancers-15-00855-f002:**
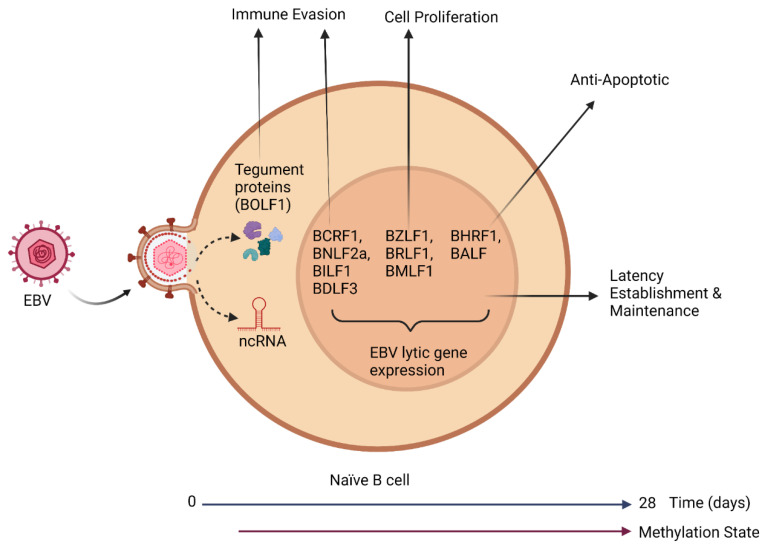
Illustration of a naïve B cell infected by EBV and the role that EBV genes expressed during the pre-latent phase have in modulating cellular functions, such as immune evasion cell proliferation and apoptosis. The pre-latent phase is reported to occur seven to twenty-eight days following infection and continues until the EBV genome is hypermethylated.

**Figure 3 cancers-15-00855-f003:**
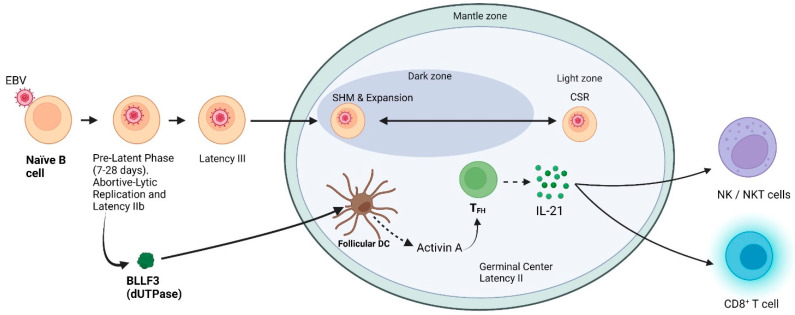
Schematic diagram showing actions of the EBV dUTPase protein contributing to the development of follicular T cells and the subsequent production of IL-21 during the pre-latent phase.

**Figure 4 cancers-15-00855-f004:**
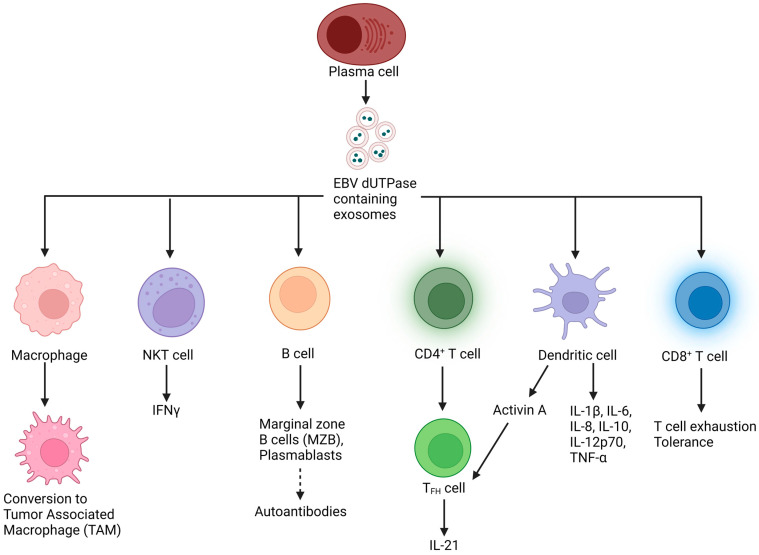
Potential roles of EBV dUTPase in modulating the TME through activation of immune cells (DCs, macrophages, NKT) and secretion of proinflammatory cytokines, promoting TAM formation, T cell exhaustion, aberrant germinal center T_FH_ function, and polyreactive antibodies production, which together can contribute to the TME and promote tumor progression. Dashed arrow represents an unproven process.

**Table 1 cancers-15-00855-t001:** Lytic EBV Genes Expressed During the Pre-Latent Phase.

Gene	Function	Phase	References
BZLF1	Transcriptional Activator	IE	[28,31,33]
BRLF1	Transcriptional Activator	IE	[33]
BMLF1	Transcriptional Activator	E	[13,35]
BCRF1	Immune Evasion	L	[28,33]
BNLF2a	Immune Evasion	E	[33]
BHRF1	BCL2 homolog	E	[13,30,33]
BALF1	BCL2 homolog	E	[30,33]
BLLF3	T_FH_	E	[37]
BILF1	Immune evasion	L	[37]
BDLF3	Immune Evasion	L	[37]
BOLF1	Immune Evasion	L	[37]

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
