# Peer review of "EBV dUTPase: A Novel Modulator of Inflammation and the Tumor Microenvironment in EBV-Associated Malignancies"

_cancers, 2023, doi:10.3390/cancers15030855_

Round 1
Reviewer 1 Report
Overall an extremely comprehensive and useful review with interesting and useful illustrations. Would be nice to know a bit more about therapeutic implications or targets that have been tried or are being tried to know how this can be used clinically.
Overall, adds nicely to the literature for a review and a few minor modifications are recommended before publication.
1. Please be very careful and ensure to spell out all abbreviations, especially for audience who do not work directly in this field. For example, figure 1, T(FH) is not described or written out.
2. Figure 1, I think you meant, "DLBCL, GC-type" not "Follicular lymphoma, GC-type"
3. Please update to reflect the new WHO 5th edition. No longer using "EBV-positive DLBCL, NOS," instead using "EBV-positive DLBCL" and importantly, some other cases of DLBCL may also be driven by EBV but classified under the new category, "Diffuse large B-cell lymphoma associated with chronic inflammation."
4. Table missing Phase for BCRF1.
Author Response
Reviewer 1
Revision Points:
- Overall, an extremely comprehensive and useful review with interesting and useful illustrations. Would be nice to know a bit more about therapeutic implications or targets that have been tried or are being tried to know how this can be used clinically.
We thank the reviewer for the positive and encouraging evaluation of our manuscript.
While EBV dUTPase is a potential target for the development of novel therapeutics, to the best of our knowledge there has only been a single study (Chem Biodiversity 2022, e200200577) addressing this possibility. The investigators reported the phytochemical dehydroevodiamine (DHE), could be a potential drug candidate to target EBV dUTPase. However, DHE targeted the enzymatic activity of the dUTPase which is not required for the immunomodulatory functions of this protein. Thus, in our opinion it would have little if any impact on the neuro-immunomodulatory properties of the dUTPase.
Please be very careful and ensure to spell out all abbreviations, especially for audience who do not work directly in this field. For example, figure 1, T(FH) is not described or written out.
We agree with the reviewer. Abbreviations have now been described in the revised manuscript (shown as track-changes). A definition of TFH has also been included in Figure 1 legend.
- Figure 1, I think you meant, "DLBCL, GC-type" not "Follicular lymphoma, GC-type". The reviewer is correct, thank you.
We agree with the reviewer’s comment. This has been corrected in Figure 1 of the revised manuscript.
- Please update to reflect the new WHO 5th edition. No longer using "EBV-positive DLBCL, NOS," instead using "EBV-positive DLBCL" and importantly, some other cases of DLBCL may also be driven by EBV but classified under the new category, "Diffuse large B-cell lymphoma associated with chronic inflammation."
This has been corrected in the revised manuscript (see line 194) as well as reference 89.
- Table 1, missing Phase for BCRF1. This has been corrected in the revised manuscript to state BCRF1 is expressed late (L).
Reviewer 2 Report
The authors performed a comprehensive review of the role of EBV early gene products in EBV lytic, latent , and abortive-lytic replication. The EBV early gene products, dUTPase encoded by BLLF3 IE, not only contributes to the establishment of latency through the production of activinA and IL-21 but it may alter the TME, promoring oncogenesis.
This provide a new aspect of the EBV dUTPase in relation to EBV-assciated lymphoma and cancers and will likely contribute to the understandng of EBV-induced tumorigenesis.
There are several points to reconsider, as described below.
This is a review containing a summary of the authors' reserach. In a study by Ariza et al (2009), the authors stated that treatment of human monocyte-derived macrophages with anti-EBV-encoded dUTPase Ab 7D6 or anti-TLR2 Ab blocked IL-6 production by EBV-encoded dUTPase. Does this anti-dUTP Ab block the establishment of latent infection in B cells? In addition, have you examined BLLF3- deficient EBV for its latent infection and/or oncogenic potential?
Table 1, The line of BLLF3: The meaning of Tfh is unclear. The authors need a comment to expalin further.
Page 4: The authors described the function of Zta, encoded by BZLF1, as a molecular switch regulating the transition between lytic and latent phases of EBV and how Zta promotes oncogenesis. In this regard, Ali-A et al (PLOS Pathog, 2022, 18 (9), e1010886) suggested that, at least epithelial cells, Rta is the dominant transactivator of early promoters and that Zta functions primarily to support DNA replication and co-activate a subset of early promoters with Rta. This study can be added to the reference.
Figure 4: The authors described signaling pathways in DC and Tfh (lines 336-338), but the effects of EBV dUTPase on other cell types are not well described. This descritpion and related references should be added in the text or figure legend. The authors should describe what the dashed arrows mean.
Author Response
Reviewer 2
We thank the reviewer for the positive and encouraging evaluation of our manuscript.
Revision points
- This is a review containing a summary of the authors' research. In a study by Ariza et al (2009), the authors stated that treatment of human monocyte-derived macrophages with anti-EBV-encoded dUTPase Ab 7D6 or anti-TLR2 Ab blocked IL-6 production by EBV- encoded dUTPase. Does this anti-dUTP Ab block the establishment of latent infection in B cells? In addition, have you examined BLLF3- deficient EBV for its latent infection and/or oncogenic potential?
These are very interesting questions which are currently unknown. These studies have not been done to the best of our knowledge.
- Table 1, The line of BLLF3: The meaning of Tfh is unclear. The authors need a comment to explain further.
We agree with the reviewer. This has been corrected and a description is now included in Figure 1 legend of the revised manuscript. Tfh in Table 1 has been changed to TFH to be consistent throughout manuscript.
- Page 4: The authors described the function of Zta, encoded by BZLF1, as a molecular switch regulating the transition between lytic and latent phases of EBV and how Zta promotes oncogenesis. In this regard, Ali-A et al (PLOS Pathog, 2022, 18 (9), e1010886) suggested that, at least epithelial cells, Rta is the dominant transactivator of early promoters and that Zta functions primarily to support DNA replication and co-activate a subset of early promoters with Rta. This study can be added to the reference.
The reviewer is correct in that Ali et al recently reported that the transcriptional activator encoded by BRLF1 (Rta) an immediate-early gene may be the dominant transactivator in epithelial cells. However, the current manuscript is written for the special issue “Study on Tumor Microenvironment in Lymphoma” and thus that is the primary focus. BRLF1 appears to have a limited effect on oncogenesis (see Rosemarie and Sugden 2020, Microorganisms 8:1824) but this is an understudied area. As such we have chosen not to add this information to the manuscript.
- Figure 4: The authors described signaling pathways in DC and Tfh (lines 336-338), but the effects of EBV dUTPase on other cell types are not well described. This description and related references should be added in the text or figure legend. The authors should describe what the dashed arrows mean.
The description of the potential effects of EBV dUTPase protein on macrophages (lines 319-321) and B-cells (lines 451-458) along with the pertinent references were included in the original manuscript.
As suggested by the reviewer, Fig 4 and legend have been revised and a definition for dashed arrow included (see below).
Figure 4. Potential roles of EBV dUTPase in modulating the TME through the activation of immune cells (DCs, macrophages, NKT) and secretion of proinflammatory cytokines, promoting TAM formation, T cell exhaustion, aberrant germinal center TFH function and polyreactive antibodies production, which together can contribute to the TME and promote tumor progression. Dashed arrow represents an unproven process.
Reviewer 3 Report
Clear and well-written. Informative in an already crowded field. Graphics are well done and aid in understanding the mechanisms you are illustrating.
Would have liked to see a bit more supporting data and elaboration on mechanisms of dUTPase oncogenicity in nasopharyngeal and gastric cancers.
Author Response
Reviewer 3
We thank the reviewer for the positive and encouraging evaluation of our manuscript.
Revision points
- Would have liked to see a bit more supporting data and elaboration on mechanisms of dUTPase oncogenicity in nasopharyngeal and gastric cancers.
This manuscript was written for the special issue “Study on Tumor Microenvironment in Lymphoma” as such the reviewer’s suggestion to discuss mechanisms of dUTPase oncogenicity in nasopharyngeal and gastric cancers while interesting are out of the scope of this review.
Reviewer 4 Report
Thanks for the opportunity to review the manuscript “EBV dUTPase: A novel modulator of inflammation and the tumor microenvironment in EBV-associated malignancies”. This article summarized the current knowledge concerning the potential roles of “abortive-lytic” replication of EBV in establishing the latent state and its contribution to the tumor microenvironment (TME) and oncogenesis.
It is well written, however, there are some issues which could be improved.
#1. The part3-6 is somewhat confusing and disorganized. The authors discuss many diseases which are related to EVB including virous lymphomas and other tumors, also discuss the modulation of the TME. I suggest the diseases should be displayed in order of disease category (lymphomas, other tumors and TME).
#2. Follicular lymphoma should be elaborated more detailed.
#3. I suggest to add figure(s) to present part3, 5, 6.
#4. Authors should check English grammar and English punctuation throughout the text.
Author Response
Reviewer 4
We thank the reviewer for the positive and encouraging evaluation of our manuscript.
#1. The part3-6 is somewhat confusing and disorganized. The authors discuss many diseases which are related to EVB including virous lymphomas and other tumors, also discuss the modulation of the TME. I suggest the diseases should be displayed in order of disease category (lymphomas, other tumors and TME).
We respectfully disagree with this comment of the reviewer. The sections are clearly indicated (3. Germinal Center Malignancies, 4. Modulation of Tumor Microenvironment, 5. Other EBV Associated Malignancies and 6. Autoimmune Disease and Lymphoma) and are displayed in a logical sequence. Thus, no changes were made in the revised manuscript.
#2. Follicular lymphoma should be elaborated more detailed.
As stated in the original submission, EBV positive follicular lymphoma is a relatively rare and poorly characterized malignancy (prevalence = 2.6%; Mackrides et al Modern Pathology 30: 519; 2017). The current consensus is that it is an understudied and a critical research gap which calls for follow-up investigations focused on examining the expression pattern of EBV lytic genes in FL patients and those undergoing transformation to an aggressive form of DLBCL. Due to lack of literature reports concerning BLLF expression in this type of lymphoma it could not be discussed in more detail.
#3. I suggest to add figure(s) to present part3, 5, 6.
We respectfully disagree with the reviewer’s comment. In our opinion Figure 2 adequately addresses section 3 and we do not believe that sections 5 or 6 require additional Figures.
#4. Authors should check English grammar and English punctuation throughout the text. This has been done as suggested by the reviewer. Please see corrections as track-changes in the revised manuscript.